# Towards LifeSpan Cognitive Systems

**Yu Wang**[1]*, **Chi Han**[2]*, **Tongtong Wu**[3]*, **Xiaoxin He**[4]*,
**Wangchunshu Zhou**[5], **Nafis Sadeq**[1], **Xiusi Chen**[2,6], **Zexue He**[1,7],
**Wei Wang**[6], **Gholamreza Haffari**[3], **Heng Ji**[2], **Julian McAuley**[1]
[1] *UCSD,* [2] *UIUC,* [3] *Monash,* [4] *NUS,* [5] *AIWaves,* [6] *UCLA,* [7] *MIT-IBM*

**Reviewed on OpenReview:** *https://openreview.net/forum?id=LZ9FmeFeLV*

## Abstract

Building a human-like system that continuously interacts with complex environments—whether simulated digital worlds or human society—presents several key challenges. Central to this is enabling continuous, high-frequency interactions, where the interactions are termed experiences. We refer to this envisioned system as the **LifeSpan Cognitive System (LSCS)**. A critical feature of LSCS is its ability to engage in incremental and rapid updates while retaining and accurately recalling past experiences. In this paper we focus on the domain of Large Language Models (LLMs), where we identify two major challenges: (1) Abstraction and Experience Merging, and (2) Long-term Retention with Accurate Recall. These properties are essential for storing new experiences, organizing past experiences, and responding to the environment in ways that leverage relevant historical data. Unlike language models with continual learning, which typically rely on large corpora for fine-tuning and focus on improving performance within specific domains or tasks, LSCS must rapidly and incrementally update with new information from its environment at a high frequency. Existing technologies with the potential of solving the above two major challenges can be classified into four classes based on a conceptual metric called **Storage Complexity**, which measures the relative space required to store past experiences. Each of these four classes of technologies has its own strengths and limitations while we argue none of them alone can achieve LSCS alone. To this end, we propose a potential instantiation for LSCS that can integrate all four classes of technologies. The new instantiation, serving as a conjecture, operates through two core processes: Absorbing Experiences and Generating Responses.

## 1 Introduction

Imagine an environment such as a simulated digital world (Park et al., 2023), a virtual world like Minecraft (Wang et al., 2023b), the Marvel Universe depicted in movies, or even the complexities of human society. Developing an intelligent system capable of engaging with such environments—interacting with its surroundings, absorbing information from experiences, self-evolving based on feedback (Zhou et al., 2024), and living through an entire lifespan, either as a real or virtual entity—remains a significant challenge. When the system inhabits the environment, it interacts with everything around it, including humans, objects, and other elements of the environment. We define these interactions as the system's **experiences**, encompassing all of the system's behaviors and the feedback it receives from the environment—whether visual, physical, linguistic, or across any other modality. This system's cognitive capabilities include the ability to perceive and interpret its surroundings, retain and recall information through memory, learn and adapt from feedback, and engage in reasoning and decision-making processes. These cognitive functions enable the system to navigate its environment intelligently and evolve over time. We denote the system with the above capabilities as the **LifeSpan Cognitive System (LSCS)**. A real-world example of LSCS is a long-lived, autonomous AI agent—such as a virtual companion or an agent operating within a simulated AI civilization. Over time,

---

*Y. Wang, C. Han, T. Wu and X. He contribute equally. Correspondence to `yuw164@ucsd.edu`.

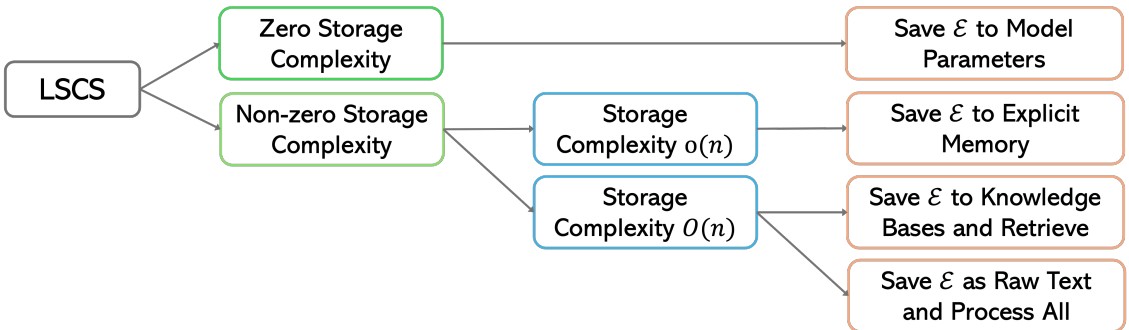

Figure 1: The technologies for constructing Lifespan Cognitive Systems (LSCS) can be broadly categorized into two principal approaches based on storage complexity. The first approach, characterized by zero storage complexity, involves **Saving $\mathcal{E}$ into Model Parameters**. The second approach, which involves non-zero storage complexity, is subdivided into methods with storage complexities of $o(n)$ and $O(n)$. The $o(n)$ methods indicate **Saving $\mathcal{E}$ into Explicit Memory**. The $O(n)$ methods are further classified based on whether the language model in the system processes the entire stored text. Suppose an additional retriever is adopted and the language model only accesses a snippet of the stored experiences. In that case, it falls under the category of **Saving $\mathcal{E}$ into Knowledge Bases for Retrieval**. The final category encompasses methods where the entire context is input into the model, classified as **Saving $\mathcal{E}$ into Raw Text and Process All**.

| Where to store | Sub-Cat | Abs & Ex-Mg | Long-Re & Ac-Re | SC |
|---|---|---|---|---|
| **Parameters** | - | ✓ | ✗ | 0 |
| **Explicit Memory** | Fixed-sized | ✓ | Partial | $O(1)$ |
| | Flexible-sized | Partial | Partial | $o(n)$ |
| **Knowledge Base** | Knowledge graph | Partial | ✓ | $O(n)$ |
| | Organized Text | | | |
| **Context** | - | ✗ | ✓ | $O(n)$ |

Table 1: In this paper, we discuss four categories of technologies with the sub-categories being sub-sections. Here "Abs & ExMg" refers to "Abstraction and Experiences Merging", "Long-Re & AcRe" means "Long-Term Retention and Accurate Recalling", and "SC" represents "Storage Complexity".

it would continuously interact with its environment, accumulate experiences, and refine its internal models. These interactions might span years, requiring the agent to recall past events and lessons learned weeks, months, or years earlier. In this paper, we mainly focus on constructing LSCS based on large language models (LLMs), as LLMs are currently the most promising direction to achieve human-like systems. Moreover, we specifically focus on text domain as most technologies discussed in this paper are constrained in text domain.

To achieve LSCS, some major challenges must be addressed: **(1) Abstraction and Experience Merging**: The system must distill experiences from its environment by extracting essential information and integrating these abstracted experiences with its existing memories. Interactions with the environment—whether through conversations (Feng et al., 2021; 2022) or visual inputs (Wang et al., 2024h)—often contain redundant information that should be filtered out before storing in memory. Once the key information is abstracted, this crucial information can be used to update the system, merging new abstracted experiences with previous ones while resolving potential conflicts. This process enables the system to acquire new skills, deepen its understanding, correct misconceptions, and continually evolve. **(2) Long-Term Retention and Accurate Recalling**: For systems that store past experiences in various forms—whether in memory, context, knowledge bases, or model weights—it is crucial to accurately recall relevant information in response to current queries from the environment. This capability necessitates the model's ability to remember events from the distant past and make informed decisions based on all past experiences. The above two properties distinguish

constructing LSCS from existing challenges such as long context problems (Wang et al., 2024d) or continual learning (Wu et al., 2024), where long context problems focus on solving a problem with extremely abundant information and continual learning mainly focuses on enabling models to capture ever-changing world knowledge (Zhang et al., 2023c; Jang et al., 2022a), facilitating self-evolution through human feedback (Tao et al., 2024), or adapting to specific domains (Ke et al., 2023; Yadav et al., 2023).

There are existing technologies that address some of the challenges described above, but each typically falls short of the other. We denote the past experiences as $\mathcal{E}$. Suppose $\mathcal{E}$ can be quantified (such as the number of tokens in the interactions between the system and the environment) and let this quantity be $n$. We then define **Storage Complexity** as a conceptual measure of the storage requirements beyond model parameters, expressed as a function of $n$, the amount of past experiences. These technologies can be categorized into four groups (illustrated in Figure 1), which will be elaborated in the subsequent sections: **Saving $\mathcal{E}$ into Model Parameters** (Section 2), **Saving $\mathcal{E}$ into Explicit Memory** (Section 3), **Saving $\mathcal{E}$ into Knowledge Bases for Retrieval** (Section 4), and **Saving $\mathcal{E}$ into Raw Text and Process All** (Section 5). We provide more details of the categorizations in Appendix A. The advantages and potential drawbacks of each approach are discussed in Section 6, with a high-level summary provided in Table 1. This table also outlines the structure of the paper, further subdividing some categories.

With the limitations of existing technologies, we argue that any category of technologies cannot achieve LSCS alone. To deal with this issue, we propose a new possible formulation for LSCS, which integrates all the approaches above. The operational design is depicted in Figure 2, where the process is split into two stages: (1) Abstraction and Merging Experiences when new experiences arise. (2) Generating Responses based on queries from the environment, ensuring long-term retention and precise recall. Each component in this design corresponds to one of the technology categories discussed earlier, and their roles are explored in more detail in Section 7.

## 2 Methods of Saving $\mathcal{E}$ into Model Parameters

The objective of integrating $\mathcal{E}$ into model parameters is allowing LLMs to continuously adapt and refine their knowledge and functionalities without repeated retraining from scratch. In the context of LSCS, these methods would store all the experiences in the model parameters. They do not require additional modules, thus the Storage Complexity is zero. The continual acquisition of experiences includes two different paths: model editing and continual learning, while continual learning can be further split into three main steps: continual pretraining, continual instruction tuning, and continual alignment. To integrate experiences into model parameters, current technologies can support injecting a large amount of domain-specific data to adapt the system via continual pre-training, or regularizing the model's behaviors via continual fine-tuning. Directly taking the various forms of inputs from the environment and storing them into model parameters are under-explored.

### 2.1 Absorbing $\mathcal{E}$ via Model Editing

Given past experiences, it is possible to construct a knowledge graph from these experiences and then convert the triplets in the graph into factual statements, which could be further injected into the model parameters via model editing (Yao et al., 2023). Here we focus on the methods that directly make edits on the model parameters, i.e., the new knowledge is saved in parameters. Methods such as ROME (Meng et al., 2022), MEMIT (Meng et al., 2023) propose a closed-form solution to edit the MLP layers, while Model-Editing-FT (Gangadhar & Stratos, 2024) fine-tunes the whole model on the factual statements that need to be injected into the model. T-Patcher (Huang et al., 2023c) and CaliNet (Dong et al., 2022) propose to store the new knowledge in additional neurons, however, storing knowledge in additional neurons may inevitably encounter the issue of ever-growing parameters in lifespan settings. As model-editing methods typically focus on storing simple factual statements in the model, they are not directly applicable in the context of LSCS. Thus adjustments are needed for the application.

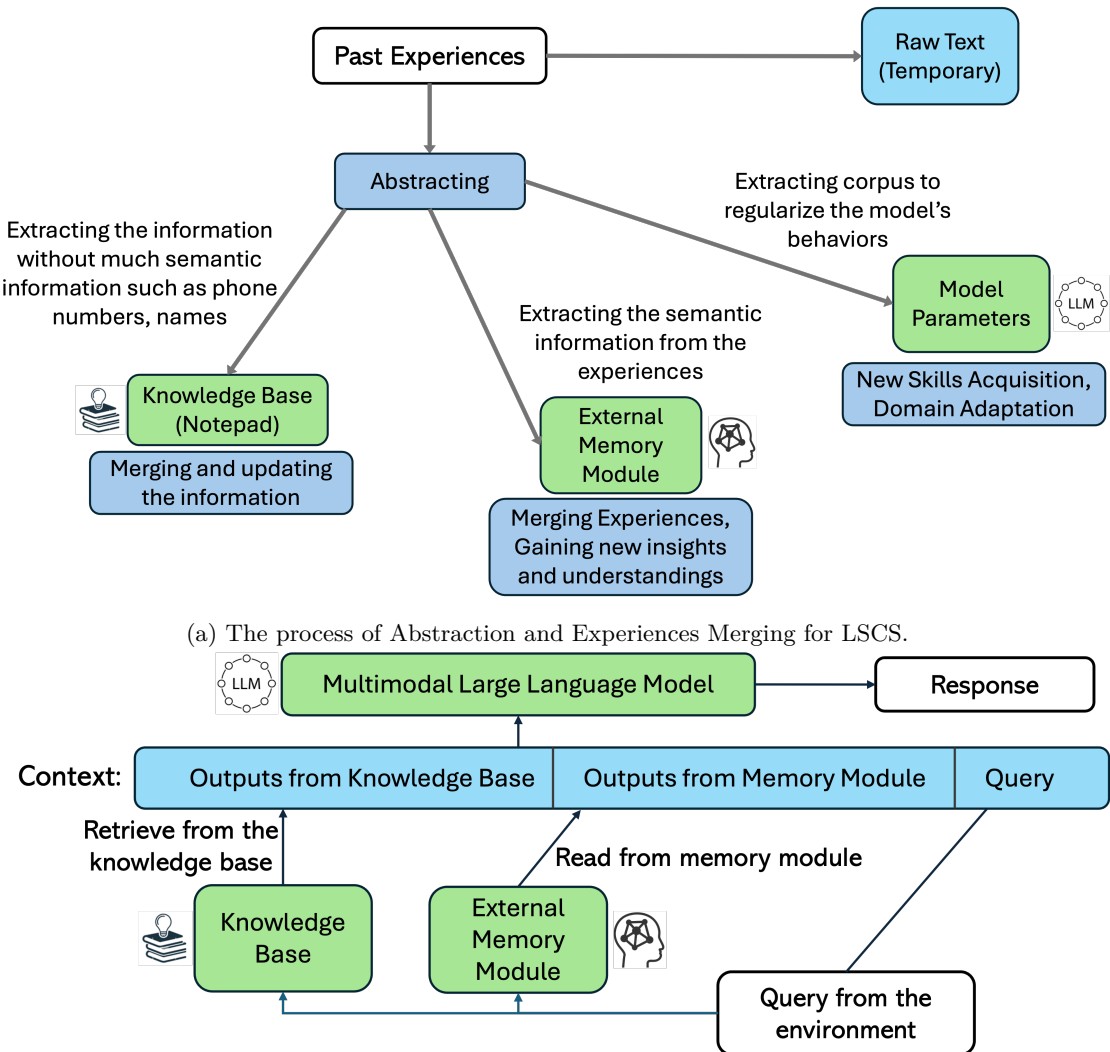

(a) The process of Abstraction and Experiences Merging for LSCS.

(b) The process of LSCS to generate responses given the query from the environment.

Figure 2: The Operating Diagram of LSCS mainly includes two parts: (a) Abstraction and Experiences Merging; (b) Generating responses according to the environmental query, where the ability of long-term retention and accurate recalling should be guaranteed. Note that we add "Notepad" after "Knowledge Base", which is meant as an analogy. See Section 7.1 for more details.

## 2.2 Absorbing $\mathcal{E}$ via Continual Learning

Continual learning is usually used in the following situations: (1) Performing real-time assimilation of dynamic data. Existing methods can incorporate information from various sources such as news (Sun et al., 2020; Yu & Ji, 2023), scholarly articles (Cossu et al., 2022), and social media (Cossu et al., 2022). Sun et al. (2020) present ERNIE 2.0, which is a continual pre-training framework that incrementally builds and learns from multiple tasks to maximize knowledge extraction from training data. (2) Injecting a large amount of knowledge into the language model. Related methods using knowledge distillation to preserve existing knowledge is proposed in DER++Buzzega et al. (2020). Jang et al. (2022b) introduce continual knowledge learning, a method for updating temporal knowledge in LLMs, reducing forgetting while acquiring new information. (3) Domain adaptation. Cossu et al. (2022) continually train the model on new data streams for both language and vision, and Ke et al. (2023) introduce a soft-masking mechanism to update language models with domain corpora, aiming to boost performance while preserving general knowledge. Various models are proposed as

the domain-adapted version of base language models such as financial domain (Xie et al., 2023), E-commerce domain (Ma et al., 2023), and academic content (Wei et al., 2023). These techniques could all be used for LSCS to adjust to the most up-to-date information. (4) Teaching the system how to speak and how to perform certain types of reasoning. To enable the models to solve novel tasks, task-incremental continual instruction tuning is proposed (Wang et al., 2024e; 2023d). These tasks could be related to mathematical problems (Azerbayev et al., 2023), calculators, search engines, and databases (Qin et al., 2023). With the rapid emergence of new tools like advanced software libraries, novel APIs, or domain-specific utilities (Liang et al., 2023; Jin et al., 2023), there is a growing need to continually update language models so they can quickly adapt and master these new tools. (5) Adapting to evolving societal values. This leads to continual alignment, where two categories exist: (i) Updating LLMs to align to changing societal values; (ii) Incorporating new demographic groups or value systems into LLMs. Some representative works include CPPO (Zhang et al., 2024b), COPF (Zhang et al., 2023a), COPR (Zhang et al., 2024a), which aim to prevent the forgetting of old policies while injecting incremental preferences objectives into the model. For LSCS, these techniques are important as the system may encounter situations where it needs to be up-to-date, learn plenty of new knowledge, adapt to a new domain if the system enters a new environment, learn a specific kind of tasks, or has to update to meet the societal values.

**Catastrophic Forgetting.** One major limitation of Saving $\mathcal{E}$ into model parameters is catastrophic forgetting, which is a general problem for methods that require updating model parameters both in model-editing (e.g. in sequential editing scenarios, only LLMs with an external module can work in sequential editing tasks such as WISE (Wang et al., 2024c), GRACE (Hartvigsen et al., 2022) while methods without an external module typically fail (Meng et al., 2022; 2023)) and continual learning. LLMs trained on different stages can encounter unconscious forgetting (Lin et al., 2023), eroding their general knowledge through continual instruction tuning. Studies (Qi et al., 2023) have shown that the behavior of safely aligned LLMs can degrade under these conditions. Some metrics are proposed in TRACE (Wang et al., 2023d) to measure the forgetting of LLMs. While we want to prevent catastrophic forgetting, we do intend to have some extent of forgetting, which is aligned with the ability of **Abstraction and Experiences Merging**. Different from unlearning part of the knowledge in the LLM which is conducted after having the undesired knowledge in the model (Wang et al., 2024g), we focus on encouraging the model to abstract the information before injecting new experiences into model parameters, where it is acceptable to discard some details. However, Abstracting experiences and discarding unimportant details are not fully explored in continual learning. It is well known that a language model cannot fully remember the training data and instead extracts inherent statistical patterns from the dataset (Carlini et al., 2023). This aligns closely with our goals in LSCS, where we want to abstract events and discard unimportant details. Intuitively, it would be beneficial to control the continual learning process such that only critical information (essential for future decision-making and answering questions) is retained in the model parameters, while less relevant knowledge is discarded. Knowledge Unlearning (Wang et al., 2024g) is related to this objective, as it focuses on removing specific knowledge from the model, which however, may negatively impact model performance (Si et al., 2023). In summary, achieving controllable continual learning is beneficial but under-explored.

## 3 Methods of Saving $\mathcal{E}$ into Explicit Memory

Different from saving $\mathcal{E}$ into model parameters, using an external memory can help store much more experiences. The techniques in this category can be split into two sub-categories: Fixed-sized Memory Module and Flexible-sized Memory Module. As the former one has a fixed size, the storage complexity is $O(1)$. As for the latter one, because there is usually some certain form of compression and forgetting mechanisms, these works mostly have a memory pool with a size that does not grow linearly with the experiences. Thus the storage complexity for past experiences is $o(n)$.

### 3.1 Methods with a fixed-sized memory module

Most existing works with fixed-sized memory modules have a small memory. Before the transformer era, plenty of works augment recurrent neural networks (RNNs) with a small memory module such as Memory Network (Weston et al., 2014) and its follow-up Sukhbaatar et al. (2015). More advanced methods

include TARDIS (Gulcehre et al., 2017), ARMIN (Li et al., 2019), Fast Weights (Ba et al., 2016), Ordered Memory (Shen et al., 2019), etc. Borrowing ideas from human memory, dual-memory system to mimic short-term and long-term memories is proposed (Bidokhti & Ghaemmaghami, 2022a;b). Later with the introduction of transformers, Memory Transformer (Burtsev & Sapunov, 2020) adds memory tokens at the beginning of the sequence to summarize the given sequence, while RMT (Bulatov et al., 2022; 2023) proposes to add memory tokens both at the beginning and the end of the sequence. Here Bulatov et al. (2023) scales the number of memory tokens to 512 which enables long-term retention of previous information. EMMA (Moro et al., 2023) has a memory pool of similar size, including both short-term and long-term memory. Then Larimar (Das et al., 2024) proposes to use a fixed-sized episodic memory which is shown to be capable of storing up to 1000 factual statements. As opposed to small fixed-sized memory, MemoryLLM (Wang et al., 2024f) introduces a memory for Llama2-7B (Touvron et al., 2023b) encapsulating 1B parameters with 7680 memory tokens per layer, enormously enlarging the memory pool. Differently, Infini-Attention (Munkhdalai et al., 2024) proposes to use linear attention to attend to the past seen tokens, where the stored matrix performs exactly like a fixed-sized memory (of size $d_{key} \times (d_{value} + 1) \times H \times L$, where $d_{key}, d_{value}, H, L$ refer to the dimension of key states, dimension of value states, number of heads and number of layers, respectively) abstracting the past tokens. These fixed-sized memory modules typically come with higher compression than flexible-sized memory modules and also requires much more training for deployment.

## 3.2 Methods with flexible-sized memory pool

The flexible-sized memory pool has various forms: (1) Hidden space, where hidden states within transformers or key-value pairs are stored in the memory pool; (2) Text-based memory, where the pool consists of textual data; and (3) Symbolic space, where the memory pool contains abstracted forms such as symbols. We describe these forms of memory pools below.

**Memory Module in Hidden Space**. Some methods introduce memory slots alongside the inputs to encode information, where the number of memory slots vary according to the length of the input document (Al Adel & Burtsev, 2021). Other techniques store key-value pairs in a memory pool for future retrieval such as KNN-LM (Khandelwal et al., 2019), Memorizing Transformer (Wu et al., 2022), LONGMEM (Wang et al., 2023c), CAMELoT (He et al., 2024b) and Memoria (Park & Bak, 2024). Here CAMELoT and Memoria include some forgetting mechanism to ensure the number of key-values pairs do not grow linearly as the input sequence becomes longer. Most recently, Memory$^3$ (Yang et al., 2024) proposes to store the knowledge from the pretraining dataset in a knowledge base within the hidden space with $1.1 \times 10^8$ text chunks with length bounded by 128 tokens. Here the number of text chunks can be easily adjusted and the number of tokens in each chunk can vary thus the size of the memory is flexible.

**Memory Module in Text Space**. Another line of work proposes to use language models as the abstractor, which inherently extracts the information from the given input and merge them with the existing memory to form a new memory. Here the memory is in the form of a summary of the past seen document. RecurrentGPT (Zhou et al., 2023), as one representative work, proposes to recurrently manage a summarization of the past context. Similarly, MemoryBank (Zhong et al., 2023) designs a hierarchical memory pool, storing (1) the raw conversation history between the user and the agent, (2) the summarization of the conversation, and (3) the summarization of the user's personalities. Here the last two summaries are in the form of text and serve as the compressed version of the past experiences. Different from RecurrentGPT which is specifically designed for long-context tasks, MemoryBank mainly focuses on conversations. There are also some forgetting mechanisms introduced in MemoryBank to mimic human memory and avoid the ever-growing problem of the memory pool. Moreover, SCM (Wang et al., 2023a) maintains the summary of each conversation and also an interaction embedding to represent the semantics of the interaction.

**Memory Module in Symbolic Space**. The memory pool in symbolic space includes database as done in ChatDB (Hu et al., 2023) where the database is updated and queried using the SQL commands generated by the fine-tuned language model. Then Voyager (Wang et al., 2023b) maintains a code base to represent the ever-growing skill library to store and update complex behaviors in Minecraft. These strategies might have the best Long-term Retention and Accuracy Recalling ability as the past experiences are integrated explicitly in the database or the code base. However, real-world experiences encountered over a lifespan may be difficult

to encode into a code base (not like the Minecraft skill set) or the database. This suggests while encoding knowledge into symbolic space could solve specific problems, its may be limited in general domains.

# 4 Methods of Saving $\mathcal{E}$ into Knowledge Bases

An effective strategy for LSCS involves creating knowledge bases from past experiences ($\mathcal{E}$) that can be accessed through retrieval-augmented generation (RAG). In this section, we discuss two major lines of methods for storing $\mathcal{E}$: (1) saving $\mathcal{E}$ as organized text; (2) saving $\mathcal{E}$ as knowledge graphs. Here saving $\mathcal{E}$ as organized text means that every detail in $\mathcal{E}$ is stored, while saving $\mathcal{E}$ as knowledge graphs involves extracting triplets from $\mathcal{E}$ and absorbing these triplets into knowledge graphs. Both approaches enable the system to maintain a repository of accumulated experiences that can be retrieved efficiently and used by LLMs during generation.

## 4.1 Saving $\mathcal{E}$ as Organized Text

The first method involves saving $\mathcal{E}$ in the form of structured or organized text. This approach is similar to traditional document-based retrieval systems, where past experiences are encoded as text documents. These documents serve as retrieval units that can be accessed when needed. In LSCS, the knowledge base is continuously updated to reflect new experiences, ensuring that the LLM can access relevant information when required. The knowledge base in this case is in the form of organized text.

**Chunking $\mathcal{E}$ into pieces.** Creating a text-based knowledge base involves chunking $\mathcal{E}$ into smaller, manageable units for retrieval. Common chunking strategies divide the text into fixed-sized segments based on token count (Guu et al., 2020; Huang et al., 2023a; Izacard et al., 2023; Siriwardhana et al., 2022). Determining the optimal chunk size is crucial for effective retrieval. Large chunks (e.g., 512 tokens) can capture more context but may include irrelevant information, making it harder to pinpoint specific answers. Small chunks (e.g., 128 tokens) are more concise but may miss important contextual details necessary for accurate retrieval. Depending on the specific dataset requirements, chunking methods range from fine-grained sentence retrieval (Cheng et al., 2023; 2024; Jiang et al., 2023b) to coarse-grained document or group of documents retrieval (Shi et al., 2023; Khattab et al., 2022; Yu et al., 2022; Jiang et al., 2024).

**Updating the text-based knowledge base.** In LSCS, the knowledge base is updated as new experiences are encountered. This process can be done in real-time or periodically, ensuring the newest experiences are incorporated. Methods such as asynchronous re-embedding and re-indexing help keep the stored experiences up-to-date (Guu et al., 2020; Huang et al., 2023a; Izacard et al., 2023; Siriwardhana et al., 2022). Open-source tools like LangChain and LlamaIndex facilitate real-time updates by continuously integrating new data into the vector database.

**Generating responses according to the knowledge base.** When a query is presented, the RAG mechanism retrieves relevant text from the knowledge base, which is then used to augment the LLM's prompt. This process allows the LLM to generate responses that reflect both the query and the relevant stored experiences. Text-based retrieval systems typically use either sparse keyword-based retrievers, such as BM25 (Robertson et al., 2009), or dense embedding-based retrievers, which capture semantic similarities between the query and the stored knowledge (Izacard et al., 2021; Ram et al., 2021). Fine-tuning these retrievers can further improve their performance in domain-specific tasks (Shi et al., 2023; Zhang et al., 2023b; Zan et al., 2022; Dai et al., 2022).

## 4.2 Storing Knowledge as Knowledge Graphs

The second method for storing $\mathcal{E}$ is using knowledge graphs, where experiences are represented as structured data (tree or graph), allowing for a more relational organization of knowledge, capturing not just the content of experiences but also the relationships between different entities and concepts. Knowledge graphs are particularly well-suited for tasks that require reasoning over connections between pieces of information.

**Creating the knowledge graph and Retrieving**. To create a knowledge graph, we first need to extract entities and relations (i.e. triplets) from $\mathcal{E}$ and then transform them into graph structures, with nodes representing entities and edges representing relationships. The constructed knowledge graphs allow for

richer retrieval possibilities compared to simple text-based storage. For instance, a system can retrieve not just isolated facts but also the relational context in which those facts exist, providing deeper insights during retrieval and generation tasks. In recent works, several methods have been proposed to improve the efficiency and accuracy of graph-based retrieval (Achiam et al., 2023; Edge et al., 2024). For instance, RAPTOR (Sarthi et al., 2024) and MemWalker (Chen et al., 2023a) utilize tree-based structures to facilitate retrieval by providing contextual information at various levels of abstraction. THEANINE (Kim et al., 2024) and AriGraph (Anokhin et al., 2024) construct memory graphs to organize memories for efficient retrieval. Additionally, KGP (Wang et al., 2023e) propose building an index across multiple documents using knowledge graphs. IIER (Guo et al., 2024) construct a chunk-interaction graph to capture the internal connections between document chunks. Inspired by the hippocampal indexing theory of human long-term memory, HippoRAG (Gutiérrez et al., 2024) employs an LLM to process passages into knowledge graph triples and leverages the Personalized PageRank algorithm to perform context-based retrieval. G-Retriever (He et al., 2024a) presents a retrieval approach for general textual graph tasks by formulating subgraph retrieval as a Prize-Collecting Steiner Tree optimization problem.

**Updating the knowledge graph**. Updating knowledge graphs in LSCS involves dynamically incorporating new nodes and edges as new experiences are acquired (Wang et al., 2023e; Guo et al., 2024; Gutiérrez et al., 2024). This ensures that the knowledge graph remains up-to-date and reflects the system's evolving understanding of its environment. Knowledge graphs can also be further linked with external data sources.

**Generating with the retrieved sub-graph**. To perform generation, Once the relevant sub-graph is retrieved, it is used to augment the LLM's prompt for generation (Hu et al., 2024; Chen et al., 2024). The structured nature of knowledge graphs provides additional benefits during this stage, as the system can leverage the explicit relationships between entities to generate more coherent and contextually informed responses. This can be particularly useful in complex reasoning tasks, where understanding the connections between concepts is as important as retrieving the correct information.

# 5 Methods of Saving $\mathcal{E}$ into Raw Text and Process All

One simple way of storing the past experiences $\mathcal{E}$ is to store all of them in the context without abstraction. In this way, the model could directly attend to these experiences when queried by the environment. As all past experiences are stored without loss of any details, the storage complexity is $O(n)$. Below we discuss the technologies that can help process the long lifespan experiences.

## 5.1 (Claimed) Length-Generalizable Architectures

Transformers Vaswani et al. (2017) is the de facto mainstream backbone architecture for most modern LLMs Achiam et al. (2023); Touvron et al. (2023a;b); Wang & Komatsuzaki (2021). To augment the self-attention layers that are position-agnostic with position information, traditional absolute positional encodings provide the absolute position information, usually with the help of a sequence of vectors called *position embeddings* (Vaswani et al., 2017; Kenton & Toutanova, 2019; Ke et al., 2020). They, however, inherently have trouble when the model encounters unseen positions in inputs longer than the training length. Inspired by this limitation, relative positional encoding techniques are proposed to depend on the attention logit function only on the relative distances between tokens instead of their absolute positions. Examples include a learned attention logit bias in T5 (Raffel et al., 2020), Transformer-XL (Dai et al., 2019), Skyformer (Chen et al., 2021), Sketching (Chen et al., 2022) and Sandwich (Chi et al., 2023), a fixed linear attention decay Press et al. (2021), and rotating query and key sequences based on distances such as RoPE (Su et al., 2021; Li et al., 2023), CAPE (Likhomanenko et al., 2021) and XPos (Sun et al., 2022; Ding et al., 2023). As a representative example widely used in multiple LLMs, RoPE (Su et al., 2021) rotates the key and query vectors based on positions before computing the inner product. Specifically, each vector $\mathbf{x}$ (either key or query) is split into pairs of elements $\{(x_0, x_1), (x_2, x_3), \cdots\}$, with each pair interpreted as a 2-dimensional vector. Another example method is AliBi (Press et al., 2021), which offsets all attention logits between tokens $i, j$ by a linear term $-m(i-j)$ and becomes $\mathbf{q}_i^\top \mathbf{k}_j - m(i-j)$. To this end, the MPT-7B codes implement an offset matrix as an additive term in attention logits. Both papers claimed to generalize to lengths longer than the training length. However, length generalization failures are still widely observed when directly applied

to off-the-shelf Transformer-based LLMs (Kaiokendev, 2023). These models also suffer from overwhelming quadratic computational complexity, and the information lost issue (Liu et al., 2024; Shang et al., 2024), which limits their deployment on practical computing devices and efficacy on extreme lengths.

In view of the limitations in the Transformer architecture, some papers revisit the concept of recurrent networks and structured state space models (SSMs) (Gu et al.) for both length-generalizable and efficient architectures. Recurrent structures usually have a linear computational complexity with respect to the sequence length, a desirable property for handling long contexts. They also maintain a bounded or sometimes fixed information bottleneck, which naturally prevents feature drift when length increases and can alleviate the generation quality degradation issue. RWKV (Peng et al.) revisit the traditional recurrent neural networks (RNNs) and demonstrates that, contrary to the common belief, a large enough RNN can exhibit impressive performance comparable to many Transformer-based LLMs. Similarly, Retnet (Sun et al., 2023) propose retentive networks, and Memba (Gu & Dao, 2023) propose to use SSMs, which combines more complicated techniques such as gating mechanism and prefix sum to improve the expressiveness and efficiency further. However, the recurrent structure also requires a delicate tradeoff between long-term and short-term memory (Gu & Dao, 2023; Beck et al., 2024), and their performance on long-context tasks, especially on lifespan data, which requires information retention on the order of years, is still under debate and investigation.

## 5.2 Length Extension Methods for Existing LLMs

Various methods have been proposed to extend the context length of existing LLMs, addressing their length limits and tackling the lifespan data better. One line of work focuses on modifying the attention mechanism without changing the model parameters. Han et al. (2024); Xiao et al. (2024) propose to modify the attention mechanism without changing the model parameters to enable LLMs to handle infinite context lengths. In particular, LM-Infinite (Han et al., 2024) extends LLMs to an extreme length of 200M tokens with $O(n)$ complexity and without degradation in perplexity metric while also showing improvements in downstream tasks. Jin et al. (2024) propose to group tokens into blocks and let each block share a relative position before applying the attention mechanism. They also demonstrate improved performance on long-context tasks, even though the grouping operation is not extendable to infinite size, which upper bounds the extension length.

Another line of work focuses on fine-tuning the model on longer texts to adapt to longer contexts. As many absolute or relative position encoding methods are trained on mathematically pre-defined periodic functions, Qu (2023); Ding et al. (2024); Liu et al. (2023); Jiang et al. (2023a) propose to apply LLMs (fine-tuned or not) on decreased period frequencies (which is equivalent to interpolating position indices) to adapt LLMs to longer sequences. This modification changes the computational features of the model, making fine-tuning necessary to adapt the model to the new context length. Some other papers finetune LLMs with designed attention patterns (Oren et al., 2024; Zhang et al., 2024c) on long contexts, using neural tangent kernel (Peng et al., 2023), or with low-rank adaptation(LoRA) (Chen et al., 2023b). Yang & Hua (2024) propose a wait-to-attend mechanism to extend length limits for memory-based Transformers. Other ways of key-value cache selection/eviction methods are investigated in Ren & Zhu (2024); Dong et al. (2024); Zhang et al. (2024f). Similarly, Huang et al. (2023b); Lee et al. (2024) tackle long context by learning to prune, select, or summarize contexts dynamically. Alternatively, context compression methods (Shao et al., 2024) propose to learn to compress long contexts into shorter embedding sequences. Some work proposes alternative position encodings (Song et al., 2023; Zhang et al., 2024e; Zhu et al., 2023) or landmark token embeddings (Luo et al., 2024) that enable extendable context limits. Compared with the other methods, especially continual learning, this category of methods needs to expand the memory continuously when the system obtains new experiences, while the former could absorb these experiences into the model parameters without expanding anything. These approaches, relying on LLMs to aggregate and process contextual information on the fly, might encounter problems in the face of mutually conflicting instructions or tasks like privacy protection and information retrieval (Wang et al., 2024a). On the other hand, cognitive systems with an awareness of the ideal situation are expected to better maintain and handle the ongoing tasks with complete interactions.

| Where to store | How Compressed ↑ (1-4) | E.`Write` ↑ (1-4) | E.`Read` ↑ (1-4) |
|:---:|:---:|:---:|:---:|
| **Parameters** | 4 | 1 | 4 |
| **Explicit Memory** | 3 | 2 | 3 |
| **Knowledge Base** | 2 | 3 | 2 |
| **Context** | 1 | 4 | 1 |

Table 2: Properties of different categories. "How Compressed" means how compressed past experiences are when stored in the corresponding form. E.`Write` and E.`Read` refer to the efficiency of `Write` process (writing the experiences into a certain form (model parameters/explicit memory/knowledge base/context)) and `Read` process (reading the experiences from a certain form), respectively. The more compressed the information is, or the higher the efficiency is, the higher score we list here.

## 6  Benefits and Limitations of the Methods in Each Category

In this section, we examine the strengths and weaknesses of the various technologies mentioned above (The strengths and weaknesses are summarized in Table 1 and Table 2).

**Saving $\mathcal{E}$ into Model Parameters**. This category includes model editing and continual learning. Current model editing techniques fall short in editing the model with life experiences which could be more complicated than factual statements, and continual learning methods still struggle with catastrophic forgetting. Despite these limitations, these methods still offer valuable contributions to LSCS, particularly when there is a need to inject substantial knowledge into the model, adjust its behavior, or update the system to reflect shifts in societal values. As the model updates its parameters with new experiences, new memories are conceptually merged with existing ones. This process leads to automatic **Abstraction and Experiences Merging**, where the raw information is not preserved in full detail, but rather abstracted and integrated into the model. Such methods face challenges with **Long-Term Retention and Accurate Recalling** due to the inherent risk of catastrophic forgetting in continual learning and failure in sequential model editing (Although some works like WISE (Wang et al., 2024c) claim to support numerous sequential edits by using external modules, these should technically fall under the category of "**Saving $\mathcal{E}$ into Explicit Memory** "). The efficiency of writing the experiences into the model parameters (E.`Write`) is low, as it involves training and back-propagation (even in model editing methods such as Meng et al. (2022; 2023), backpropagation is needed). In contrast, the efficiency of reading the experiences (E.`Read`) is high, as the generation speed remains unaffected by updates while the effects of the experiences can still be reflected in generation.

**Saving $\mathcal{E}$ into Explicit Memory**. Memory-based methods show promise as they allow the model to access information from the distant past by storing $\mathcal{E}$ in memory, without significantly increasing inference costs as history lengthens. These methods (1) often involve some form of abstraction, resulting in the inability to predict exact answers when queried on specific questions, as only the abstracted information from the experiences is stored. (2) may be hard to enable long-term retention as there is either explicit forgetting (Wang et al., 2024f; Zhong et al., 2023) or implicit forgetting (Zhou et al., 2023). One possible way to mitigate the forgetting issue is to enlarge the memory pool as studied in MemoryLLM (Wang et al., 2024f). However, the utilization of the memory pool in MemoryLLM may not be optimal, as it can only achieve around 40 steps (approximately 20k input length) of updates without completely forgetting the earlier knowledge. Despite the current limitations of memory-based methods, we believe memory-based methods are still an important part of achieving a powerful LSCS. In LSCS, one important ability is **Abstraction and Experiences Merging**. In memory-based methods, there are certain forms of compression when writing the experiences into the memory, which aligns with the idea of abstraction (abstracting the incoming knowledge). However, experiences merging is still under-explored in memory-based methods. Though there are some preliminary explorations (He et al., 2024b), how to merge the memories, especially in the hidden space is extremely hard and there are barely any works capable of merging memories in a way that similar memories are put together and even lead to new understandings instead of simply concatenating all memories. As for the ability of **Long-Term Retention and Accurate Recalling**, memory-based methods are inherently not able to recall accurate information as the compression and forgetting mechanisms are part of the design of

the memory module, which means some detailed information is destined to be missing. However they can still achieve a long-term retention where the major contour of events could be maintained with details forgotten. The efficiency of the `Write` process is higher than saving $\mathcal{E}$ into model parameters as it usually requires a forward pass to convert the text into some certain space, while the efficiency of the `Read` process is lower than saving $\mathcal{E}$ into model parameters as the LLM needs to read from the memory module and put this extracted information in the context to process.

**Saving $\mathcal{E}$ into Knowledge Bases for Retrieval**. storing $\mathcal{E}$ in knowledge bases is essential for achieving LSCS, enabling the system to accumulate vast amounts of knowledge over time. The vast amounts of knowledge in the past can be stored as organized text or knowledge graphs, where RAG methods can be used to retrieve the relevant knowledge from the organized text or knowledge graphs (Gutiérrez et al., 2024; Sarthi et al., 2024) to augment the generation process. While text-based storage excels at handling large volumes of general information, knowledge graphs offer more structured and relational insights. Together, these methods allow LSCS to retrieve and utilize past experiences effectively, adapting to new situations and challenges based on accumulated knowledge. This category includes methods that store experiences as raw text in a knowledge base or create structured representations, like graphs, by extracting triplets from the input data. When extracting triplets, some detailed information that cannot be represented by factual triplets is inevitably lost, and the nodes and edges in the knowledge graph can be updated to resolve conflicts and merge similar relations. However, these abstractions are fairly naive and only simple conflicts and merging can be handled. As for the knowledge base in the form of organized text, the process usually needs to abstract some knowledge from the experiences to create index such as the embedding of summary of the experiences, where the experiences merging could be even harder than using knowledge graph. Thus we say this category of methods involve Partial **Abstraction and Experiences Merging**. These methods offer strong **Long-Term Retention and Accurate Recalling**, as the knowledge base can store extensive information. The efficiency of the `Write` process (**E.Write**) is higher than the above two categories, as it involves processing the input to construct structured representations which may be faster than converting all text into some other spaces. The efficiency of `Read` process (**E.Read**) is lower than the above two, as it requires retrieving and integrating relevant information from the knowledge base. In comparison, Saving $\mathcal{E}$ into model parameters does not require retrieval, and saving $\mathcal{E}$ into explicit memory could potentially skip the need of retrieval such as MemoryLLM (Wang et al., 2024f).

**Saving $\mathcal{E}$ into Raw Text and Process All** While saving $\mathcal{E}$ as the raw text and putting it into context is conceptually straightforward, it faces significant practical challenges. For instance, humans, are estimated to speak an average of 16k words per day (Mehl et al., 2007), totaling hundreds of millions of words over a lifetime (Brandreth, 1980). Consequently, current LLMs usually cannot process all past experiences. Even methods that claim to handle infinitely long contexts struggle with effectively recalling important knowledge from the past (Gu & Dao, 2023; Sun et al., 2024). Despite extensive efforts to extend these length limits, obstacles such as computational inefficiency and information forgetting remain unresolved Liu et al. (2024), preventing these approaches from being a fundamental solution to LSCS. As for the property **Abstraction and Experiences Merging** mentioned in Section 1, this line of methods does not involve any abstraction or experiences merging. However, as all the past experiences are stored explicitly in the context, this category of methods is capable of achieving **Long-Term Retention and Accurate Recalling**. The `Write` process (**E.Write**) is highly efficient, as it merely involves concatenating new experiences to the existing data. However, the `Read` process (**E.Read**) is less efficient, as it requires the language model to process all stored experiences, leading to increased computational demands.

## 7 Our Proposed Instantiation of LSCS

Existing technologies individually address some aspects of the challenges associated with **Abstracting and Merging Experiences** and **Long-Term Retention and Accurate Recall**, but none can handle both sets of issues comprehensively. To address these limitations, we propose a new instantiation that integrates and combines these methods to create a more robust solution. As depicted in Figure 2, our instantiation operates in two key phases: absorbing experiences and generating responses to environmental queries, with the detailed descriptions shown below:

### 7.1 Absorbing Experiences

The experience absorption process within LSCS occurs with multiple levels of abstraction. This allows the system to retain both raw details and high-level summaries of experiences, which are processed and stored in several levels (corresponding to Figure 2a):

**Raw Experience Storage**: The latest experiences are captured in their raw textual form, preserving details without loss. This information is stored in the system's memory as a direct record of events, which is accessible for immediate recall.

**Non-semantic Information Storage**: Certain types of data, such as phone numbers or some hard-to-remember names, are challenging for typical memory models to store effectively. These pieces of non-semantic information, which lack comprehensive contextual meaning but are critical for precision, are stored in a dedicated knowledge base. The knowledge base acts as a repository for factual, easily forgotten information that LSCS can access as needed. Just as humans may use notebooks to store specific pieces of information they cannot reliably and easily retain in their memory (e.g., phone numbers, exact names, addresses), an LSCS could have an external, easily accessible storage component serving a similar role. For instance, frequently updated, high-fidelity information like contact details, passcodes, or historical records (generally "verbatim" facts that are cumbersome to store in model parameters) would be stored in such a dedicated external knowledge base, thus we add the term "Notepad" in Figure 2a to highlight the role of the knowledge base.

**Semantic Information Abstraction**: Experiences are also abstracted to capture their core meaning or "essence," such as a general impression of a person, an emotional response to an event, or a rough memorization of some events that happened before. These abstracted memories, which may lack fine details, are stored in the external memory module. This module allows for memory consolidation and combination, leading to the creation of new insights and deeper understandings through a process akin to memory integration and fusion. The distinction between semantic information and non-semantic information is that non-semantic information is generally much harder to remember for humans. Humans can memorize what happened in the movie (which falls under the semantic information category) after watching it once and can recall the plot after a long time but may struggle to remember some detailed information such as phone numbers, specific names that do not have any semantic meaning (we call them non-semantic information). Thus for our instantiation, we also handle these two types of information differently.

**Deeply Encoded Information in Model Parameters**: The highest level of abstraction occurs when repeated experiences are encoded directly into the model's parameters, leading to the formation of long-term behaviors and habits. This mirrors how humans internalize routines or adapt to new environments over time. In LSCS, these deeply embedded patterns represent highly compressed forms of knowledge that are retained without needing further memory resources.

As experiences are abstracted through these levels, the degree of compression increases, transitioning from raw text to structured knowledge, and eventually to internalized model parameters. During this process, conflicts between new and existing knowledge may inevitably arise. Ideally, a memory model designed to resemble human memory could effectively merge similar experiences over time, while a dynamically updated knowledge graph could help resolve conflicts and integrate new information seamlessly.

### 7.2 Generating Responses to Environmental Queries

When faced with an environmental query, LSCS gathers relevant information from all of its modules to generate a response. To begin with, the query is integrated into the context, forming the basis for the system's reasoning. Then related information is extracted with the following processes (see Figure 2b):

**Retrieval from Knowledge Base**: If factual or non-semantic information is required, the system queries the knowledge base for relevant data, which is then added to the context. This ensures that precise, detail-oriented information is readily available when needed.

**Abstracted Semantic Information Retrieval**: With the environmental query, the system reads the external memory module to extract the relevant semantic information which is usually in the hidden space. The extracted hidden states are also put into the context.

**Response Generation**: With all necessary inputs in place, the large language model (potentially a multimodal one as the experiences can include images, speech, videos as well (Wang et al., 2024h)). takes the context as the input and generate a response according to the context and the model parameters. This step includes reading the extracted information from the knowledge base, the memory module, and the model parameters to provide a complete, context-aware response to the environmental query.

### 7.3 Summary of the Proposed Instantiation

In the above-mentioned processes, both the knowledge base and memory module are utilized, allowing us to adopt the technologies described in **Saving $\mathcal{E}$ into Knowledge Bases for Retrieval** and **Saving $\mathcal{E}$ into Explicit Memory**. Since the system incorporates a large language model, the techniques outlined in **Saving $\mathcal{E}$ into Model Parameters** are also applicable. During response generation, the model must handle extensive contextual inputs, which can be long enough to require the large language model to have long-context processing capabilities. By leveraging these technologies, we can enable **Abstraction and Experience Merging**, where the knowledge base, external memory module, and model parameters can all store abstracted versions of experiences. Then with the presence of both the knowledge base and external memory module, the system is capable of achieving **Long-Term Retention and Accurate Recall** via querying these modules. We would like to highlight that the current instantiation of LSCS still remains conceptual and does not represent a fully implemented system. While we do want to propose a system that can actually work, there exist some challenges, illustrated in Appendix B.

## 8 Conclusion

In this paper, we propose the concept of a **LifeSpan Cognitive System** (LSCS), which aims to manage rapid, incremental learning while retaining the ability to recall previous experiences. To achieve LSCS, we argue there are two major challenges: (1) Abstraction and Experiences Merging, (2) Long-Term Retention and Accurate Recalling. Existing technologies have the potential to partially solve these challenges. Based on our defined metric, **Storage Complexity**, for saving past experiences ($\mathcal{E}$), we categorize existing technologies into four distinct classes. We summarize various technologies within each category and discuss the strengths and weaknesses of the methods in each category. We argue that achieving an LSCS requires a nuanced approach integrating multiple strategies to address the above two challenges. To this end, we propose a conceptual instantiation for LSCS. We hope to provide insights into LSCS and encourage further efforts.

## Acknowledgement

We thank the reviewers and the action editor for their valuable suggestions and comments. This research is based upon work supported by U.S. DARPA ECOLE Program No. HR00112390060. The views and conclusions contained herein are those of the authors and should not be interpreted as necessarily representing the official policies, either expressed or implied, of DARPA, or the U.S. Government. The U.S. Government is authorized to reproduce and distribute reprints for governmental purposes notwithstanding any copyright annotation therein.

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

# A   Additional Details of Method Classification

Here we mainly discuss why we state that saving $\mathcal{E}$ into knowledge base has $O(n)$ storage complexity and saving $\mathcal{E}$ into memories has $o(n)$ complexity. We define Storage Complexity as a conceptual measure of how the amount of stored information grows with the number of past experiences (denoted by $n$). For example, suppose the system interacts daily with a user for one year, accumulating 365 days of conversation ($n = 365$). In a knowledge-based method that stores each day's data separately, the storage may grow linearly with the number of experiences, $O(n)$. If each day's interaction is stored verbatim, the total storage after 365 days is approximately 365 conversation records. In contrast, memory-based methods might employ a forgetting mechanism. For instance, if the system only stores a summarized version of past interactions, it may reduce each day's information by a certain factor. As $n$ grows large, the total amount of stored data could become sub-linear, reflecting that older interactions are increasingly compressed or partially discarded. For example, after the first few weeks, only a summary of each earlier conversation might remain. Thus, over a long period, you might not need to store all 365 full conversations; instead, you store a few full recent ones and short summaries of older ones, resulting in less than a strictly linear increase in total storage.

# B   Challenges of Implementing our Proposed Instantiation

Two major challenges are described below:

**Limited Real-World Implementation of Memory-Based Methods**: Existing memory-oriented methods for GPT-4 level large language models (LLMs) largely are essentially texts rather than true "hidden-space" memory. Methods that have memory in hidden space, such as MemoryLLM (Wang et al., 2024f), InfiniteAttention (Munkhdalai et al., 2024), Memory[3] (Yang et al., 2024) illustrate early attempts in this direction. However, these methods have only been successfully applied to relatively small models (up to a few billion parameters) (Wang et al., 2024f) and often lack open-source implementations (Munkhdalai et al., 2024; Yang et al., 2024), making it difficult to scale and integrate with state-of-the-art large LLMs. In contrast, RAG methods can easily be applied on GPT4-level models. We can (1) Create an LSCS based on small models so that memory and RAG can both be incorporated, but these models may have limited capacities (For instance, RULER [4] shows that Llama-3.1-8B only has 32k effective context window.), which might make it hard to create a truly practical LSCS. (2) Ideally if we have a large memory language model where we can introduce RAG, then the built LSCS can be much stronger but as we said the "large memory language model" currently does not exist.

**Lack of Appropriate Long-Term Benchmarks**: Current benchmarks, such as $\infty$Bench (Zhang et al., 2024d) or Loong Bench (Wang et al., 2024b), are still at 200k tokens level, and state-of-the-art models can often solve these tasks by naively fitting all relevant context into their windows (for instance, Gemini-1.5 pro has 1M context window). In contrast, LSCS targets much longer time horizons—akin to a system's entire lifespan—and we currently lack datasets and benchmarks that reflect these extreme scales. As a result, evaluating and validating the proposed LSCS on real-world or lifespan-scale scenarios is not yet feasible.

Despite these limitations, we view LSCS as a forward-looking framework. As larger memory-equipped models, better open-source implementations, and more extensive benchmarks emerge, the LSCS concept could transition from speculation to practical realization.

Although integrating different categories of methods might be challenging, we would like to mention that a promising direction is to use a model that supports both memory tokens and RAG in a unified manner. For instance, MemoryLLM (Wang et al., 2024f) incorporates up to 12,800 memory tokens per layer and a generation context window of 2,048 tokens. Scaling this approach—e.g., to 96k memory tokens per layer and 32k context window—could enable a model to process vast amounts of stored knowledge within its hidden space. Under such a scenario, RAG techniques could then retrieve and feed external knowledge (e.g., from a notepad-like structure) back into the model, creating a seamless interplay between internally stored representations and external references. While fine-tuning such a system remains a challenge, our view is that this would be an infrequent event. In rare cases where fine-tuning is needed, one could preserve the original training and instruction datasets, mixing them into the new training set to regularize and prevent catastrophic forgetting.

