# OpenReview forum: "Towards LifeSpan Cognitive Systems"
_TMLR — Accepted by TMLR_

### Review · Reviewer_cJPq · 2024-12-04

**Summary Of Contributions:**

The paper analyzes LLM models in the lifelong learning context with two challenges, “abstraction and experience merging”, and “long-term retention with accurate recall.” To do so, authors first categorize three different ways to learn experience, task and finally propose the future form of lifespan cognitive system (LSCS).

**Audience:**

Yes

**Broader Impact Concerns:**

I do not have any broader impact concerns.

**Claims And Evidence:**

No

**Requested Changes:**

1. I think the paragraph in Section 2 can be a subsection for consistency with other sections, and the numbering in paragraph “Absorbing $\epsilon$ via Continual Learning” can be italicized to distinguish other texts.

2. It is difficult to know whether the higher number is better in Table 2 although the authors mentioned it in the last sentence of the caption. I would recommend using the arrow in the table header to visualize it explicitly.

3. There are a few formatting errors. Here are some examples.

(1). Citep and citet are misused:

- citet->citep : scholarly articles Cossu et al. (2022), and social media Cossu et al. (2022). Sun et al. (2020)

 - domain-specific utilities Liang et al. (2023); Jin et al. (2023),
- COPR Zhang et al. (2024a),
- follow-up Sukhbaatar et al. (2015).

(2). When the citation written by multiple authors is used as subject, the verb should be plural form:

-  Ke et al. (2023) introduces -> introduce.
- Sun et al. (2020) presents -> present.

(3). Long-term Retention and Accurate Recalling -> Long-Term Retention and Accurate Recalling

(4). Figure 2 caption, (1), (2) -> (a), (b) following subcaption.

(5). Year is missing for the citation “Hongye Jin, Xiaotian Han, Jingfeng Yang, Zhimeng Jiang, Zirui Liu, Chia-Yuan Chang, Huiyuan Chen, and Xia Hu. Llm maybe longlm: Selfextend llm context window without tuning. In Forty-first International Conference on Machine Learning.“

4. The first few sentences in the second paragraph of Section 6 are redundant.

**Strengths And Weaknesses:**

I consider this paper as a survey paper although the authors propose the future direction of the lifespan cognitive system (LSCS)

[Strength]
1. The paper is generally well written and conducted an extensive survey on LLM related to continual learning, and memory retrieval.

2. It seems that the categorization of existing is intuitive and valid. However, at the same time, categorization in Sections 2 and 3 slightly overlapped with standard continual learning categories (e.g., [1])


[Weakness]
Overall Comment: the claims and analysis are not well supported by citation or explanation, especially the 'new' paradigm proposal.

1. First and foremost, the scope of the paper is slightly mismatched between the abstract/introduction and the following contents. In the beginning, the paper describes the general paradigm of the LifeSpan Cognitive System (LSCS) but from Section 2, the paper only limits their scope to LLM. This can be simply resolved by mentioning their scope in both the abstract and introduction.

2. If we consider the general tasks not limited to LLM, several statements could be wrong. For example, the limitation of Absorbing $\epsilon$ via Model Editing is the network can be ever-growing. However, DER++ [2] employs knowledge distillation to retain the network size after expansion.

3. Similarly, the below statement could be a hasty generalization considering non-LLM literature.
> Compared with the other methods, especially continual learning, this category of methods needs to expand the memory continuously when the system obtains new experiences, while continual learning could absorb these experiences into the model parameters without expanding anything.

4. Moreover, it is difficult to agree with the statement that `continual learning necessitates
constructing datasets for retraining.` considering regularization-based continual learning such as EWC [3] and LwF [4] which can be applicable to any task.


5. What does “notepad” in Figure 2 and (Claimed) in Section 5.1 mean?

6. There is no definition for n in big-O and little-o notation and I am wondering why “Save $\epsilon$ to Explicit Memory” is o(n) while “Save $\epsilon$ Knowledge Bases and Retrieve” is O(n).

7. There is no citation for Updating the knowledge graph and Generating with the retrieved sub-graph paragraph although they do not look like the authors’ new proposal.

8. Some claim needs to be supported by citation. For example, the authors claim that `the recurrent structure also requires a delicate tradeoff between long-term and short-term memory`. Although I agree with this claim, I wonder whether any reader can directly understand the reason for this statement considering the paragraph that explains SSM.
Moreover, a statement, `Whether stored as organized text or knowledge graphs, the information can be retrieved efficiently through RAG methods to inform the generation process.` should also be supported by citation or be considered as a claim with supporting sentences.

9. The proposed paradigm of LSCS is not well supported. Why do the authors think phone numbers, for example, are challenging for typical memory models to store effectively? If I understand correctly, the authors try to argue that these types of data need to be memorized as is. If so, I would recommend changing the texts to be more clear. Likewise, why abstracted memories should be stored in the external memory module is not discussed. I highly recommend describing in more detail why the suggested paradigm should be like that among various design choices.

10. I personally wonder whether ‘paradigm’ is the correct term and think that the proposed paradigm, a mixture of multiple components is not surprising. However, since it is a personal opinion, I did not count them during the review.

[References]

[1] Wang, Liyuan, et al. "A comprehensive survey of continual learning: theory, method, and application." IEEE Transactions on Pattern Analysis and Machine Intelligence (2024).

[2] Buzzega, Pietro, et al. "Dark experience for general continual learning: a strong, simple baseline." Advances in neural information processing systems 33 (2020): 15920-15930.

[3] Kirkpatrick, James, et al. "Overcoming catastrophic forgetting in neural networks." Proceedings of the national academy of sciences 114.13 (2017): 3521-3526.

[4] Li, Zhizhong, and Derek Hoiem. "Learning without forgetting." IEEE transactions on pattern analysis and machine intelligence 40.12 (2017): 2935-2947.

---

> ### Author Response · Authors · 2024-12-14
> **Rebuttal to Reviewer cJPq (Part 1/3)**
>
> We thank the reviewer for their careful reading of our manuscript and their valuable suggestions. We have addressed all the points raised and will incorporate these clarifications and improvements in the final version of the paper. Our responses are organized by the reviewer’s comments:
>
> ### Categorization in Sections 2 and Sections 3
> We acknowledge the reviewer’s observation regarding potential overlap with standard continual learning categories as presented in [1]. However, we would like to clarify the differences in scope and focus. While [1] discusses `memory stability` within the context of catastrophic forgetting, its emphasis lies on preserving previously learned tasks within the model parameters. In contrast, our framework concentrates on methods that leverage external memory, expanding beyond the confines of catastrophic forgetting to consider a broader range of memory-based technologies. Moreover, [1] subdivides the field of continual learning into multiple categories, treating it as the primary subject of investigation. In our work, continual learning is itself just one sub-category within a larger landscape of techniques aimed at building LifeSpan Cognitive Systems (LSCS). Our primary focus is on the overarching paradigm of LSCS, which includes but is not limited to continual learning approaches. Thus, while there may be some conceptual resemblance, our categorization addresses a more expansive problem space and employs continual learning as one component among many technologies that contribute to long-term, dynamic cognition in LLMs.
>
> ### Scope Clarification & General Focus
> **[W1] & [W3]**: We acknowledge the confusion regarding the scope of LSCS. While we have introduced the concept of a LifeSpan Cognitive System (LSCS) in general terms, our paper focuses specifically on LLM-based systems as an initial case study. We will revise the abstract and introduction to clearly state that, although LSCS could theoretically be implemented using various architectures and modalities, the present work primarily examines LSCS in the context of large language models (LLMs). This clarification should better align the initial motivation in the abstract with the main content of the paper.
>
> ### Categorization and Comparisons with Non-LLM Literature
> **[W2]**: We apologize for the confusion when describing the limitations of certain model-editing methods. Our intent was not to imply that all model-editing or continual learning methods require ever-growing parameters. Rather, we highlighted that some specific methods (e.g., T-Patcher (Huang et al., 2023c) and CaliNet (Dong et al., 2022)) store new knowledge in additional neurons, potentially causing parameter growth. We will clarify in the revision that this limitation pertains to particular instances, not the entire category. Additionally, we will incorporate DER++ [2] as an example of a continual learning method that uses knowledge distillation to incorporate new information.
>
> ### Addressing Claims About Continual Learning and Dataset Construction
> **[W4]**: As for the statement `continual learning necessitates constructing datasets for retraining`, We recognize that our wording on “retraining” was ambiguous. Here `retraining` should be replaced with `training`. Our core point was that continual learning methods often require learning from new data (e.g., new tasks, environments, or domains), which can be viewed as “constructing” or incorporating new datasets over time. Methods such as EWC [3] and LwF [4] aim to mitigate catastrophic forgetting, but they still rely on the availability of new data or tasks. We will clarify that “constructing datasets” refers to the natural process of collecting or introducing new data for continual learning rather than a separate or burdensome process. We will rewrite this section to emphasize that while continual learning methods do not necessarily require retraining from scratch, they do need access to new data as the system encounters new experiences.
>
> ### Explanation of the “Notepad” Concept
> **[W5]**: The term “notepad” is meant as an analogy. Just as humans may use notebooks to store specific pieces of information they cannot reliably and easily retain in their biological memory (e.g., phone numbers, exact names, addresses), a LSCS could have an external, easily accessible storage component serving a similar role. For instance, frequently updated, high-fidelity information like contact details, passcodes, or historical records — generally “verbatim” facts that are cumbersome to store in model parameters — would be stored in such a dedicated external memory. We will add a clear explanation of this analogy in both Figure 2’s caption and Section 7.

---

> > ### Author Response · Authors · 2024-12-14
> > **Rebuttal to Reviewer cJPq (Part 2/3)**
> >
> > ### Complexity Notation and Examples
> > **[W6]**: We apologize for not defining $n$ explicitly. We will clarify in the revised manuscript that $n$ represents the quantity of past experiences (e.g., number of interaction steps, documents read, or tasks encountered over the system’s lifespan). We define Storage Complexity as a conceptual measure of how the amount of stored information grows with the number of past experiences (denoted by n). For example, suppose the system interacts daily with a user for one year, accumulating ~365 days of conversation (n=365). In a knowledge-base method that stores each day’s data separately, the storage may grow linearly with the number of experiences, O(n). If each day’s interaction is stored verbatim, the total storage after 365 days is approximately 365 conversation records. In contrast, memory-based methods might employ a forgetting mechanism. For instance, if the system only stores a summarized version of past interactions, it may reduce each day’s information by a certain factor. As n grows large, the total amount of stored data could become sublinear, reflecting that older interactions are increasingly compressed or partially discarded. For example, after the first few weeks, only a short summary of each earlier conversation might remain. Thus, over a long period, you might not need to store all 365 full conversations; instead, you store a few full recent ones and short summaries of older ones, resulting in less than a strictly linear increase in total storage. We will include such concrete examples and clarify the definitions in the revision.
> >
> > ### Citations and Support for Claims
> > **[W7]**: The methods mentioned in “Creating the knowledge graph and Retrieving” are also the methods that involve updating the knowledge graph and generating with the retrieved sub-graph. We will clarify this in our paper and also put the citations in these two paragraphs.
> >
> > **[W8]**: For the claim `the recurrent structure also requires a delicate tradeoff between long-term and short-term memory`, our statement is more like a rational analysis. The key idea is that if the system’s representational capacity (an “information bottleneck”) remains fixed, but the context it needs to handle keeps expanding, then the system must decide how to allocate its limited representational dimensions. This limitation is true for models such as Mamba, xLSTM and so on. We will add these papers as the references for this claim. For the second claim `Whether stored as organized text or knowledge graphs, the information can be retrieved efficiently through RAG methods to inform the generation process`, we were trying to say **The vast amounts of knowledge in the past can be stored as organized text or knowledge graphs, where RAG methods can be used to retrieve the relevant knowledge from the organized text or knowledge graphs**, which is actually supported by any RAG paper. We will revise this statement and add some citations like HippoRAG and RAPTOR here.

---

> > > ### Author Response · Authors · 2024-12-14
> > > **Rebuttal to Reviewer cJPq (Part 3/3)**
> > >
> > > ### Clarification on Why Certain Types of Data Are Challenging
> > > **[W9]** We agree that the text regarding “challenging types of data” (like phone numbers) needs more clarity. Our intention was to highlight that some data must be recalled verbatim (e.g., exact numeric or symbolic values), making it difficult for the model to rely solely on internal parameterization. Such data do not easily benefit from compression or abstraction and are prone to being “forgotten” or altered over time. Storing them externally (e.g., in a knowledge base or “notepad”) ensures reliable recall. We will rewrite the corresponding section to explain that highly specific, “exact-match” information is best handled by external storage to prevent distortion and ensure accuracy over a long lifespan. We will further justify why abstractions (such as summarized facts or learned patterns) might be integrated into the model’s parameters, while verbatim data can be retained reliably and at scale in external memory.
> > >
> > > ### Clarification on the word `Paradigm`:
> > > **[W10]** We appreciate the reviewer’s personal opinion regarding our use of the word “paradigm.” While we understand that the term may feel broad or potentially overused, our intention is to emphasize the conceptual shift toward integrating multiple, complementary components into a cohesive LifeSpan Cognitive System (LSCS). Although the combination of different techniques might seem unsurprising in isolation, we believe the underlying principle of continuously adapting and scaling memory, retrieval, and reasoning mechanisms over a system’s entire lifespan represents a meaningful conceptual framework. Still, we acknowledge that the choice of terminology can be subjective, and we will consider clarifying our wording or providing more concrete definitions in the final version.
> > >
> > > ### Additional Clarifications & Formatting (Requested Changes)
> > > 1. We will convert the paragraphs in Section 2 into subsections. And we will italicize the numberings to distinguish from other texts.
> > > 2. We will add the arrows in the table header for better visualization.
> > > 3. (1) We will carefully check the paper to correct the citep and citet errors. (2) We will check if the papers we cite are written by multiple authors to correct the forms of the verbs. (3) We will fix the word "Long-term" to "Long-Term". (4) We will change (1) (2) in the caption to (a) (b) to match the figure. (5) We will check the references to make sure the years are included.
> > > 4. In Section 6, second paragraph, we will shorten some of the sentences to avoid conveying repeated information and improve readability.
> > >
> > > ### References
> > >
> > > [1] Wang, Liyuan, et al. "A comprehensive survey of continual learning: theory, method, and application." IEEE Transactions on Pattern Analysis and Machine Intelligence (2024).
> > >
> > > [2] Buzzega, Pietro, et al. "Dark experience for general continual learning: a strong, simple baseline." Advances in neural information processing systems 33 (2020): 15920-15930.
> > >
> > > [3] Kirkpatrick, James, et al. "Overcoming catastrophic forgetting in neural networks." Proceedings of the national academy of sciences 114.13 (2017): 3521-3526.
> > >
> > > [4] Li, Zhizhong, and Derek Hoiem. "Learning without forgetting." IEEE transactions on pattern analysis and machine intelligence 40.12 (2017): 2935-2947.

---

> > ### Comment · Reviewer_cJPq · 2024-12-16
> >
> > I appreciate the authors for the detailed response.
> >
> > **Regarding the authors’ response for W4**
> >
> > The authors are currently swapping the causation and the result. Continual learning refers to the situation in which a new task/data comes in while the previous task/data is usually not accessible. However, the authors argue that continual learning needs new data or tasks. This is a fallacy of putting the cart before the horse.
> >
> > **Regarding o(n)**
> >
> > I am still not fully convinced of the time complexity of knowledge bases and retrieve o(n). If you use small-o notation, authors must guarantee that the complexity must be smaller than n strictly but the authors’ explanation is not rigorous enough.
> >
> >
> > For other responses, I will wait for the authors’ revision as the authors can freely revise during the discussion period.

---

> ### Author Response · Authors · 2024-12-16
> **Response to cJPq**
>
> We sincerely thank the reviewer for the prompt response.
>
> As the reviewer mentioned, "Continual learning refers to the situation in which a new task/data comes in while the previous task/data is usually not accessible." So how do you perform continual learning? On the new task/data that comes. We are saying when you get the data, the fine-tuning process needs to be performed on the data, not that "In order to do continual learning (for no reason), we need some kind of data". Thus the **reason** to perform continual learning is always to **satisfy some goal such as adapting to a new domain**, then we need to collect some data for continual learning. The reason or the goal is never **to do continual learning**. We apologize for the confusion introduced in our manuscript and we will revise our manuscript carefully to avoid such confusion.
>
> We would like to mention that the complexity of knowledge bases is O(n), and the memory mechanisms have the complexity o(n). This is to say that knowledge bases will expand linearly as new knowledge continuously comes in but memory should either have sublinear complexity or constant complexity. If a method claims to be memory-based but has O(n) complexity, it should actually be put it under RAG category. We define this metric in order to categorize the methods, rather than saying existing methods that claim to be in the categories RAG or Memory-based methods will all have the storage complexity we proposed. In other words, we are not summarizing existing methods and then proposing the metric, instead, we propose the metric and then categorize the existing methods.
>
> We will revise the manuscript and put the new manuscript here as soon as possible.

---

### Review · Reviewer_xvSy · 2024-12-07

**Summary Of Contributions:**

This paper proposes the concept of a LifeSpan Cognitive System (LSCS), an intelligent agent designed to continuously interact with its environment, accumulate experiences, and apply them to future behaviors. The authors identify two main challenges: effectively abstracting and merging experiences into coherent memory structures, and maintaining long-term retention with accurate recall. The paper surveys existing approaches along a Storage Complexity metric, categorizing methods based on how they store experiences: in model parameters, external memory, knowledge bases, or raw experience storage. Finding that no single method fully achieves LSCS capabilities, the authors propose integrating multiple techniques, combining abstraction through memory and knowledge bases with large language models and selective retrieval.

**Audience:**

Yes

**Claims And Evidence:**

Yes

**Requested Changes:**

It would make this work stronger if the author could make the implementation to be specific not just high-level description and include at small-scale experiment demonstrating the feasibility of the proposed paradigm. For example, simulate a simplified LSCS scenario and show how incorporating a memory module plus a knowledge base leads to improved long-term task performance compared to a baseline.

**Strengths And Weaknesses:**

Strengths:

(1) The paper clearly illustrates the goal of LSCS and identifies two core challenges that go beyond standard continual learning or long-context problems. This innovative idea gives a broader perspective on what a “human-like” cognitive system might be in the real world.

(2) It provides a helpful taxonomy of existing methods based on “Storage Complexity” that gives a structured understanding of the solution space and trade-offs among them.

(3) The paper proposed a conceptual paradigm that integrates multiple strategies from each category to achieve the 2 argued challenges, providing valuable insights to future work.


Weaknesses:

(1) Although the paper proposes an integrative paradigm, it does not present experiments or quantitative results. Without empirical demonstrations, it is unclear how the combined approach would perform in practice, or how to benchmark progress towards LSCS.

(2) Most of the discussion remains theoretical. It is challenging to translate these concepts into concrete implementations. The paper would be stronger if it included preliminary prototypes or at least detailed case studies illustrating how the proposed paradigm might be realized.

(3) While the paper recognizes storage complexity as a critical metric, it provides only qualitative insights into efficiency. There is less discussion on the computational or memory overheads of these integrated systems, and no concrete strategies are given for scaling them to real-world datasets.

(4) While the author argues the proposed lifespan cognitive "system", it primarily focuses on languages as input without discussing the other input data modalities such as the vision for LVM and also the multiple modality (LMM).

(5) Although experience merging and conflict resolution are identified as key challenges, the paper provides relatively sparse detail on how such processes might be automated or improved. The high-level approach is clear, but the operational details and algorithms are not well specified.

---

> ### Author Response · Authors · 2024-12-15
> **Official Rebuttal to Reviewer xvSy**
>
> We thank the reviewer for their thoughtful feedback and for recognizing the broader perspective and conceptual framework we introduce with the LifeSpan Cognitive System (LSCS). We appreciate the insights and will incorporate them to improve and refine our work.
>
> We understand that the reviewer’s primary concern is the lack of practical implementation and empirical evidence. However, we wish to emphasize that our paper is intended as a position paper. Our goal is to define the conceptual challenges of LSCS, highlight key research directions, and inspire the community to develop the necessary benchmarks, tools, and methodologies that will make LSCS a practical reality. Given the complexity of the problem and the current limitations of available tools and datasets, providing a fully implemented and empirically validated solution at this stage is not feasible.
>
>
> Below, we respond to each identified weakness in more detail:
>
> - **[W1] Lack of Empirical Demonstrations and Benchmarking**:
>     We acknowledge the value of empirical evidence. However, LSCS aims at handling much longer time horizons than existing benchmarks (e.g., ∞Bench, Loong Bench, or even models with million-token contexts like Gemini-1.5 pro). Truly “lifespan-scale” interactions would span far beyond a few million tokens and demand sophisticated strategies for abstraction, retrieval, and forgetting. As such, there are currently no datasets or evaluation frameworks that adequately reflect the extended, evolving contexts LSCS envisions. Without such benchmarks, it is challenging to provide quantitative metrics or conclusive empirical results. Our current contribution is to identify this gap and encourage the community to create the resources needed for future empirical exploration.
>
> -  **[W2] Bridging Concepts to Concrete Implementations**:
>    Our paper is conceptual by design. Rather than presenting a tested methodology, we aim to define what a LSCS could look like and why existing methods are insufficient. While it may be possible to prototype partial solutions, the absence of suitable benchmarks and stable, large-scale memory-based methods for massive LLMs makes it difficult to validate and refine such prototypes. By highlighting these limitations, we hope to motivate further work toward developing the infrastructure and resources required to move from concept to implementation.
>
> - **[W3] Efficiency and Scalability Considerations**:
>     Our four-category taxonomy provides a high-level framework, illustrating how various approaches differ in storage complexity and resource requirements. Given the range of existing methods - each with distinct computation efficiencies and storage complexities, it might be overly exhaustive to list all methods with detailed, quantified efficiency and storage complexity, which also requires a well-curated benchmark for evaluation and might be far beyond our current paper. Instead, we propose the conceptual differences between different categories, which should be sufficient enough to categorize and distinguish different methods.
>
> - **[W4]: Considering Ohter Modalities**:
>   We agree that a truly “human-like” cognitive system should integrate multiple modalities beyond language (e.g., vision, audio). However, current research and tooling are heavily optimized for language-only scenarios. To avoid confusion, we will clarify in the revised manuscript that we focus on LLM-based scenarios as an initial case. We hope that the conceptual groundwork laid here can eventually guide the extension of LSCS principles to multimodal systems once robust multimodal benchmarks and models are readily available.
>
> - **[W5]: How to Perform Experience Merging and Conflict Resolution**:
>   These are indeed central challenges for LSCS. While we cannot yet provide detailed algorithms, we suggest that improved memory mechanisms, and integrated knowledge graphs could help. For example, a memory model that more closely resembles human memory might effectively merge similar experiences over time, while a knowledge graph could be dynamically updated to resolve conflicts and integrate new information. We will clarify in the revised version how these ideas form a foundation for future work on automated knowledge integration and conflict resolution.

---

### Review · Reviewer_La4p · 2024-12-11

**Summary Of Contributions:**

The authors present a new framework, called LifeSpan Cognitive System (LSCS), for building systems capable of continuous, high-frequency interaction with complex environments with emphasis on rapid, incremental learning and long-term retention with accurate recall of past experiences. The authors propose combining multiple existing technologies, categorized by Storage Complexity, into a new paradigm that unites these approaches.

**Audience:**

Yes

**Claims And Evidence:**

Yes

**Requested Changes:**

- Can you provide concrete examples of real-world applications for the proposed LSCS framework? What would be the computational and financial costs associated with maintaining continuous updates, high-frequency interactions, and large-scale memory systems?
- Section 2: "It is well known that a language model cannot fully remember the training data." Can you please elaborate on it further in connection to learning/unlearning? How does this align with the goals of LSCS?

Some minor writing/presentation-related issues:
- Table 1 column name referred to in the caption doesn't match
- Section 3.1: "plenty of works argument ..."

**Strengths And Weaknesses:**

Strengths:
- The paper is overall well-written and the authors presented ideas in an organized manner.
- Comprehensive details are provided for individual groups of storage requirements. The concept of Storage Complexity offers an insightful way to evaluate and compare methods for storing past experiences.
- The LSCS framework encapsulates critical aspects of the human cognitive system, addressing essential elements for continual learning and effective interaction with the environment.

Weaknesses:
- The proposed LSCS, while ambitious, still remains conceptual and speculative, with no practical implementation details provided.
- Integrating approaches with different strengths and limitations poses significant design challenges. The paper does not entirely clarify how such approaches can be combined seamlessly. I think that the existing methods may not yet be fully capable of addressing all aspects of the stated goals, especially since many modules, such as continual pre-training and fine-tuning large language models (LLMs), remain active and highly challenging areas of research.

---

> ### Author Response · Authors · 2024-12-14
> **Rebuttal to Reviewer La4p (Part 1/2)**
>
> We thank the reviewer for their thoughtful and constructive feedback. We address each of the concerns and requested changes as follows:
>
> - **[W1] Conceptual Nature and Implementation Details**:
>
>     We acknowledge that the current form of the LifeSpan Cognitive System (LSCS) is primarily conceptual and does not present a fully implemented system. Two key challenges hinder practical implementation at this stage:
>
>     **Limited Real-World Implementations of Memory-Based Models:**
>     Existing memory-oriented methods for GPT-4 level large language models (LLMs) largely are essentially texts rather than true “hidden-space” memory. Methods that have memory in hidden space, such as MemoryLLM [1], InfiniteAttention [2], and Memory³ [3] illustrate early attempts in this direction. However, these methods have only been successfully applied to relatively small models (up to a few billion parameters) [1] and often lack open-source implementations ([2, 3]), making it difficult to scale and integrate with state-of-the-art large LLMs.
>     In contrast, RAG methods can easily be applied on GPT4-level models.
>     We can (1) Create an LSCS based on small models so that memory and RAG can both be incorporated, but these models may have limited capacities (For instance, RULER [4] shows that Llama-3.1-8B only has 32k effective context window.), which might make it hard to create a truly practical LSCS. (2) Ideally if we have a large memory language model where we can introduce RAG, then the built LSCS can be much stronger but as we said the "large memory language model" currently does not exist.
>
>     **Lack of Appropriate Long-Term Benchmarks:**
>     Current benchmarks, such as $\infty$Bench or Loong Bench, are still at 200k tokens level, and state-of-the-art models can often solve these tasks by naively fitting all relevant context into their windows (for instance, Gemini-1.5 pro has 1M context window). In contrast, LSCS targets much longer time horizons—akin to a system’s entire lifespan—and we currently lack datasets and benchmarks that reflect these extreme scales. As a result, evaluating and validating the proposed LSCS on real-world or lifespan-scale scenarios is not yet feasible.
>
>     Despite these limitations, we view LSCS as a forward-looking framework. As larger memory-equipped models, better open-source implementations, and more extensive benchmarks emerge, the LSCS concept could transition from speculation to practical realization.
>
> - **[W2] Integrating Different Approaches Seamlessly**:
>     We appreciate the reviewer’s concern regarding the complexity of integrating diverse technologies. A promising direction is to use a model that supports both memory tokens and RAG in a unified manner. For instance, MemoryLLM [1] incorporates up to 12,800 memory tokens per layer and a generation context window of 2,048 tokens. Scaling this approach—e.g., to 96k memory tokens per layer and 32k context window—could enable a model to process vast amounts of stored knowledge within its hidden space.
>
>     Under such a scenario, RAG techniques could then retrieve and feed external knowledge (e.g., from a notepad-like structure) back into the model, creating a seamless interplay between internally stored representations and external references. While fine-tuning such a system remains a challenge, our view is that this would be an infrequent event. In rare cases where fine-tuning is needed, one could preserve the original training and instruction datasets, mixing them into the new training set to regularize and prevent catastrophic forgetting.
>
> **References:**
>
> [1] MemoryLLM: Towards Self-Updatable Large Language Models.
> [2] Leave No Context Behind: Efficient Infinite Context Transformers with Infini-Attention.
> [3] Memory³: Language Modeling with Explicit Memory.
> [4] RULER: What's the Real Context Size of Your Long-Context Language Models?
> [5] $\infty$Bench: Extending Long Context Evaluation Beyond 100K Tokens
> [6] Self-Updatable Large Language Models with Parameter Integration.

---

> > ### Author Response · Authors · 2024-12-14
> > **Rebuttal to Reviewer La4p (Part 2/2)**
> >
> > **Requested Changes:**
> >
> > - **Concrete Real-World Applications of LSCS & Associated Costs:** A real-world example that demonstrates the utility of LSCS is a long-lived, autonomous AI agent—such as a virtual companion or an agent operating within a simulated AI civilization. Over time, it would continuously interact with its environment, accumulate experiences, and refine its internal models. These interactions might span years, requiring the agent to recall past events and lessons learned weeks, months, or even years earlier.
> >
> >     From a computational and financial perspective, if we have a memory system, it should be easily updated. As proposed in MemoryLLM[1], ideally, the update should be efficient if the memory update only contains a forward pass. Similarly, to update the knowledge base (possibly in the form of a database), we are likely to ask our model to call some functions to update. As for the continual learning part, this should be deemed an infrequent procedure as it might be much more resource-demanding than others. We will add these clarifications to our paper.
> >
> >
> > - **Elaboration on “Language Model Cannot Fully Remember Training Data” about LSCS:**
> >   Continual fine-tuning alone does not guarantee perfect recall of previously seen information, as shown in [6]. For the task context question answering, finetuning the model on the context will not introduce the knowledge of the context into the model, leading to the inability to answer the related questions [6]. In addition, catastrophic forgetting is a known challenge: when updated with new data, LLMs may lose some previously acquired knowledge. LSCS aims to mitigate this by providing a structured, external memory that can be a stable repository of past experiences. In doing so, LSCS aligns with the long-term goal of enabling models to learn, unlearn, and relearn information more effectively, continuously updating their knowledge without sacrificing what they have learned before. We will add these elaborations into our paper.
> >
> > **Minor Writing/Presentation Corrections:**
> >
> > We will ensure that the column names mentioned in Table 1’s caption match those in the table.
> > In Section 3.1, we will correct the phrase “plenty of works argument” to “plenty of works augment”
> >
> > **References:**
> >
> > [1] MemoryLLM: Towards Self-Updatable Large Language Models.
> > [2] Leave No Context Behind: Efficient Infinite Context Transformers with Infini-Attention.
> > [3] Memory³: Language Modeling with Explicit Memory.
> > [4] RULER: What's the Real Context Size of Your Long-Context Language Models?
> > [5] $\infty$Bench: Extending Long Context Evaluation Beyond 100K Tokens
> > [6] Self-Updatable Large Language Models with Parameter Integration.

---

### Author Response · Authors · 2024-12-22
**Cover Letter of New Revision**

**Dear Editors and Reviewers,**

We sincerely thank you for your thoughtful and constructive feedback on our manuscript. We greatly appreciate the time and effort you have dedicated to reviewing our work, and your comments have been instrumental in improving the quality and clarity of our paper.

We have carefully addressed all the points raised in your reviews and incorporated your invaluable suggestions into our revised manuscript. Below, we summarize the key revisions, all of which are highlighted in blue in the updated manuscript for your convenience:

### Major Revisions

1. **Real-World Example**: We have included a real-world example in the first paragraph of the Introduction.
2. **Domain Clarification**: We explicitly clarify in the first paragraph of the Introduction that our paper focuses on LLMs within the text domain.
3. **Explicit Definition of $n$**: The definition of $n$ has been added at the end of page 2.
4. **Details of Categorizations**: We provide detailed explanations about RAG methods having $O(n)$ storage complexity and memory-based methods having $o(n)$ storage complexity in Appendix A, with a reference added at the beginning of page 2.
5. **Model Editing Clarifications**: Additional clarifications have been included in Section 2.1 to avoid ambiguity.
6. **Inclusion of DER++**: We now include a citation for DER++ in Section 2.2.
7. **Description of Notepad in Figure 2**: Descriptions have been added to the caption of Figure 2 and elaborated in Section 7.1.
8. **Clarifications on Memory and Knowledge Unlearning**: We expand on the discussion of "LLMs cannot fully remember training data" and "how LSCS is related to knowledge unlearning" in Section 2.2, within the paragraph on catastrophic forgetting.
9. **Citations in Section 4.2**: Additional citations and supporting evidence are included in the paragraphs **Updating the knowledge graph** and **Generating with the retrieved sub-graph** in Section 4.2.
10. **Rephrasing Redundant Sentences**: We have rephrased sentences in the second paragraph of Section 6 for better readability.
11. **Clarifications on Knowledge Base Retrieval**: Additional clarifications have been added to Section 6 regarding "Saving $E$ into Knowledge Bases for Retrieval."
12. **Semantic vs. Non-Semantic Information**: Further descriptions in Section 7.1 clarify the need for a knowledge base as a "notepad" and explain how information is classified into semantic and non-semantic categories.
13. **Conflict Resolution**: We elaborate on how to resolve conflicts at the end of Section 7.1.
14. **LSCS Challenges**: The challenges of implementing LSCS are outlined in Appendix B, and additional descriptions are included at the end of Section 7.3.

### Minor Revisions

1. Corrected all issues with `citep` and `citet` formatting.
2. Fixed grammatical errors, particularly with plural forms of verbs after papers with multiple authors.
3. Added an upward arrow in Table 1.
4. Reorganized paragraphs in Section 2 into subsections.
5. Included citations for Mamba and xLSTM following the statement: "The recurrent structure also requires a delicate tradeoff between long-term and short-term memory" in Section 5.1.

We are sincerely grateful for your invaluable feedback, which has significantly enhanced the quality of our paper. We hope that the revised manuscript meets your expectations, and we look forward to your further insights.

Thank you once again for your guidance and support.

---

> ### Comment · Reviewer_cJPq · 2024-12-25
>
> Thank you for updating the manuscript.
> One minor point: the font of O(n) in L6 in Appendix A is different from all others.
> Also, it would be nicer to mention LLM in the abstract so that readers expect the main domain of this paper.

---

> > ### Author Response · Authors · 2024-12-25
> > **Additional Revisions**
> >
> > Dear Reviewer cJPq,
> >
> > Thank you for pointing these out! We have added the description "In this paper we focus onthe domain of Large Language Models (LLMs), ... ", and fixed the font issue in the appendix.
> >
> > Best,
> > Authors

---

### Decision · Action_Editor_mKxq · 2025-01-08

**Recommendation:** Accept as is

**Comment:**

The reviewers generally found that the paper met the standards of claims and audience. However, the reviewers and I agree that the impact of the paper may be limited by its lack of in-depth development of the framework or methods of evaluation, or any concrete implementation thereof.

**Audience:**

The framework proposed by this paper may be of some interest to researchers in continual or lifelong learning.

**Claims And Evidence:**

The revised version of the paper does a better job of distinguishing between claims and conjectures or speculation. I believe it satisfies the standards of the journal (in part because it is relatively light on concrete claims).